# Design and Feasibility Study of a Leg-exoskeleton Assistive Wheelchair Robot with Tests on Gluteus Medius Muscles

**DOI:** 10.3390/s19030548

**Published:** 2019-01-28

**Authors:** Gao Huang, Marco Ceccarelli, Qiang Huang, Weimin Zhang, Zhangguo Yu, Xuechao Chen, Jingeng Mai

**Affiliations:** 1Beijing Advanced Innovation Center for Intelligent Robots and Systems, Beijing Institute of Technology, Beijing, 100081, China; huanggao@bit.edu.cn (G.H.); ceccarelli@unicas.it (M.C.); qhuang@bit.edu.cn (Q.H.); yuzg@bit.edu.cn (Z.J.); chenxuechao@bit.edu.cn (X.C.); 2Intelligent Robot Institute, School of Mechatronical Engineering, Beijing Institute of Technology, Beijing, 100081, China; 3Key Laboratory of Biomimetic Robots and Systems, Beijing Institute of Technology, Ministry of Education, Beijing, 100081, China; 4LARM: Laboratory of Robotics and Mechatronics, University of Cassino and South Latium, Cassino, 03043, Italy; 5The Robotics Research Group, College of Engineering, Peking University, Beijing, 100871, China; Jingengmai@pku.edu.cn

**Keywords:** EMG signal, master-slave control, muscle exercises, pedal-actuated wheelchair, assistive robots

## Abstract

The muscles of the lower limbs directly influence leg motion, therefore, lower limb muscle exercise is important for persons living with lower limb disabilities. This paper presents a medical assistive robot with leg exoskeletons for locomotion and leg muscle exercises. It also presents a novel pedal-cycling actuation method with a crank-rocker mechanism. The mechanism is driven by a single motor with a mechanical structure that ensures user safety. A control system is designed based on a master-slave control with sensor fusion method. Here, the intended motion of the user is detected by pedal-based force sensors and is then used in combination with joystick movements as control signals for leg-exoskeleton and wheelchair motions. Experimental data is presented and then analyzed to determine robotic motion characteristics as well as the assistance efficiency with attached electromyogram (EMG) sensors. A typical muscle EMG signal analysis shows that the exercise efficiency for EMG activated amplitudes of the gluteus medius muscles approximates a walking at speed of 3 m/s when cycling at different speeds (i.e., from 16 to 80 r/min) in a wheelchair. As such, the present wheelchair robot is a good candidate for enabling effective gluteus medius muscle exercises for persons living with gluteus medius muscle disabilities.

## 1. Introduction

Elderly people and stroke patients with hemiplegia often suffer from muscle weakness and gait disorders [1], which can result in a significant burden to their families and society in general. The number of lower limb disabilities due to old age or stroke hemiplegia is gradually increasing [2]. Moreover, muscle atrophy (and even festering) that results from prolonged periods of muscle inactivity can adversely affect general health and even lead to amputation [3]. Successful healthcare intervention is not only dependent on a clinician’s capability and experience, but also on the availability and adequacy of medical instruments and assistive devices. Due to the wide range of (daily life) applications, technical aids and assistive devices for the elderly and people living with severe motor disabilities are gaining increasing attention as the world population ages. As such, medical mechatronics is an important emerging technology in the healthcare industry [4]. A key research goal in this field is to allow people living with lower limb disabilities to experience better mobility and locomotivity in daily activities. The majority of assistive devices (or robots) have been developed to assist patients in hospital-based rehabilitation training. However, given the increase in the global elderly (aging) population, assistive devices are increasingly required for in-house rehabilitation as well [5].

Previous research shows that patients with Trendelenburg gait and post Spinal Cord Injury (SCI) can lose ‘smooth’ joint movement [6]. For Trendelenburg gait patients, the loss is due to action deficiency on one side of the hip abductor muscles (gluteus medius) [7,8]. For SCI patients, the loss is due to an inability to control thigh muscle flexibility. As such, a robot that could provide rehabilitation assistance for the abovementioned muscles would be beneficial. The wheelchair is useful for persons who have lost the ability to independently walk; it also reduces the influence of a user’s weight when utilized for rehabilitation activities. Although there are many types of wheelchairs available in the marketplace, the majority of these machines are used for travelling short distances and use hand-powered movement and control mechanisms [9]. In general, traditional wheelchair design is convenient for user-based operations and is suitable for users with lower limb amputation. However, these wheelchairs are fully manual; as such, that they are not suitable for users who lack sufficient strength for manual propulsion. While electric-powered wheelchairs supply a substantial amount of the energy required for movement (i.e., accommodate users who lack sufficient strength for manual propulsion), nevertheless, they do not provide opportunities for leg exercise. As such, wheelchairs can inadvertently contribute to muscle atrophy, thereby adversely affecting the health of the user.

Pedaling motions are not only common in cycling sports, but are also widely used in the rehabilitation of impaired muscles [10,11]. A clinical study has verified the effectiveness of cycling as a rehabilitative exercise [12]. There are several commercially available rehabilitation devices that employ the cycling motion for lower limb exercise, including NuStep [13], THERA-Vital [14], MOTOmed [15] and RehaMove [16]. These devices provide passive, motor-assisted or active-passive motions to users through training modes that are selected based on muscle strength [17]. These devices are especially suitable for people with limited mobility and those who are ‘wheelchair-bound’ [18]. While these devices benefit targeted leg rehabilitation, they do not promote user mobility, which can result in user boredom and, ultimately, a reluctance to engage with this kind of device.

Previous research shows the convenience of combining a wheelchair with leg exercises for locomotion and rehabilitation [19]. Numerous studies have focused on wheelchair-based leg cycling exercises. She et. al. proposed a three-wheel electric cart with a mounted pedal unit to enable leg exercises [20]; the cart used a bilateral master-slave control method under an impedance model to control the cycling motion [21]. While the proposed design is a good solution for users to engage in wheelchair-based leg exercises, there are no force sensors to detect the intended motion of the user (“user motion intention”). Also, motion control requires the user to manually (i.e., use his or her foot) depress the pedal to engage wheelchair movement. This action is generally uncomfortable for the user and is especially unsuitable for persons who require significant mental preparation to engage in simple motions. Furthermore, passive leg motion is difficult to accurately engage under the proposed control method. Other research has focused on Functional Electrical Stimulation (FES) to stimulate specific lower limb muscles for recumbent cycling motion [22,23,24]. Lo et al. proposed an FES-based cycling wheelchair for stroke patients engaged in the recovery stage [25]. Clinical evaluation results showed that the proposed wheelchair was easier to use and more efficient than a manual wheelchair. Tohoku University (Japan) [26] developed Profhand, a commercial cycling wheelchair, which has now been used in numerous studies [27,28,29,30,31]. Hirata et al. [27,28,29] attached a continuously variable transmission (CVT) and servo brake to control movement of the cycling wheelchair. Here, the speed of the cycling motion determines the speed of the wheelchair due to the mechanical connection to the pedal. While this is a viable solution for users with manual strength, it is not suitable for users who lack sufficient strength for manual propulsion due to the absence of motor assistance. A similar design was proposed by Kaisumi et al.; here, a servo motor and brake were added to a cycling wheelchair [30,31]. This modification resulted in better movement across a wide range of slopes. 

Under the abovementioned solutions, wheelchair and cycling motions are fixed because they use a chain transmission between the two motions; as such, proportional adjustments for the two speeds are not possible. Furthermore, there is no mechanism to assist the user to move his or her legs in the right trajectory when engaging in the cycling motion. As such, even simple propulsion can result in swerving and loss of control, both of which can adversely affect general safety and rehabilitation for the user.

This paper presents a cycling wheelchair with leg exoskeletons that enables free movement and leg cycling to address the limitations of the current cycling wheelchair and leg rehabilitation devices. General locomotion and exercise are correlated by user motion intention and can be conveniently and voluntarily adjusted by the designated controls. The exoskeletons are designed to guide and assist leg motion while maintaining user safety. Previous work proposed a leg exoskeleton cycling-actuated wheelchair robot with a master-slave control to provide locomotion and rehabilitation functions [32,33,34,35,36] using a crank-rocker mechanism. However, experimentation showed that leg exoskeleton motion was difficult to control when the motor was affixed to the rocker [30] because motion passing through the ‘dead point’ of the crank was user-assisted. In addition, the pedaling force used for motion control changed during the cycling [33]. Previous work showed that the wheelchair scheme with the leg cycling exoskeleton was convenient for lower limb muscle exercises in a wheelchair [32,33,34]. This paper investigates the main deficiencies of the original design then presents an improved mechanical design and control system to deliver better operational experience for the user. For the mechanical design, an adjustable hip mechanism is presented for a comfortable user experience. The redesigned system is predicated on the user requirement to both get on and get off the wheelchair. Additional experimental results for the abductor muscles are discussed to verify the effectiveness of the proposed wheelchair robot. Electromyography (EMG) signals are recorded from the hip and thigh muscles under different experimental conditions to analyze the effectiveness of the exercise. 

The remainder of this paper is organized as follows: firstly, the design requirements are presented and a novel conceptual design and improved mechanical design of the wheelchair robot are described. Secondly, the master-slave control method with a control design is introduced and the coordinated motion control of the wheelchair and leg exoskeletons are described. Experimental results are described and analyzed to demonstrate the feasibility of the control method. The results show that the proposed wheelchair robot is a viable healthcare mechatronic intervention for locations with sufficient space to operate the exercise device and that it is suitable for lower limb exercise and assistive locomotion devices. Finally, the conclusion is presented along with an outline for future work to optimize the present locomotion and rehabilitation exercise method.

## 2. Conceptual Design

The wheelchair robot is suitable for the environment where the majority of movement is short-distance. As such, the mechatronic design should facilitate general operations. In addition to mobility, safety is another primary consideration for the conceptual design of the robot. The robot should be easy to operate with suitable control mechanisms for the elderly and persons who lack sufficient strength for manual propulsion.

A user-oriented pedal-actuated exoskeletal wheelchair robot for rehabilitation and locomotion that ensures user safety was proposed in [32]. To meet leg muscles exercise requirements, a crank-rock mechanism was affixed on the wheelchair to enable the leg cycling motion when the wheelchair was active. This solution requires only a minor modification to the wheelchair robot in terms of mechanical parts. The modification has a fixed cycling motion trajectory with the crank and rocker lengths set to ensure user safety and ease of control. Theoretically, in the crank-rocker mechanism, when the rocker acts as the driving link, then there are two ‘dead spots’ within a single working cycle; also, controlling the motor rotation speed to produce continuous crank rotation is difficult [37]. Consequently, to simplify the control method in the present adaption, the exoskeleton motor is relocated to the crank location (from the rocker location) in the crank-rocker mechanism. Here, the continuous motor rotation in the crank location realizes exoskeleton motion in the crank, knee and hip locations. Now, leg exercise control concentrates on speed and force, while motion direction can be ignored, thereby simplifying the control.

Figure 1 shows the proposed conceptual design of a wheelchair robot with leg-exoskeletons. To address the limitations of wheelchairs that are currently available in the marketplace (i.e., locomotion functions with no assistance for leg exercises), two cycling motion are combined with a wheelchair and a crank-rocker exoskeleton mechanism is then applied to simplify lower limb joint motions. A cycling motion is also incorporated into the design to exercise the leg muscles. Figure 1a, dotted line, shows the user foot trajectory (which is unique to each user) and the related motion pathway; it is safe for persons who lack sufficient strength for manual propulsion. Figure 1b shows the conceptual design in terms of degrees of freedom (DOF). There are three active DOFs: one for leg exoskeleton motion and two for wheelchair motion. There are another nine passive DOFs: six for leg motion assistance (on the exoskeleton) and three for wheelchair motion assistance. For the leg exoskeletons, the DOFs are related to the hip, knee and ankle joints. In addition to the nine passive DOFs, the hip joint connections between the exoskeleton and wheelchair are adjustable per user preference, as shown in the green rectangular area in Figure 1b,. The conceptual design is user-focused such that it addresses user comfort, rehabilitation environment usage, and locomotion and exercise requirements.

## 3. Mechanical Design and Control

Figure 2 shows mechanical improvements to the previous design that allow users to simultaneously engage in locomotion and leg motion exercises. In the present adaption, six passive DOFs (on the exoskeleton) are designed to coordinate motion with user hip, knee and ankle joints. The passive joints are driven by a crank pedal with a single motor, which provides passive or active power for the exoskeletons and legs of the user when required. The output torque of the serve-motor for the leg exoskeletons is 13.2 Nm. The rotation motion from the motor to the crank pedal is transmitted by the chain with a transmission ratio of 1:1 per the input torque requirement of the motor. 

The length and fixed position of the exoskeleton linkages can be adjusted per height of the user. Figure 2 shows the hip joint adjustment mechanism. The exoskeleton attachments to the wheelchair board can be both horizontally and vertically adjusted i.e., they can be adjusted forwards and backwards, as shown in Figure 2b, yellow arrow, and upwards and downwards, as shown in Figure 2b, green arrow. The adjustable attachment mechanism provides a more comfortable user experience while also satisfying exercise requirements at different positions. This mechanism is a distinct feature of the present design based on user experiments with the most recent prototype. Figure 2b shows the adjustable area, which is specifically designed for an average Asian user [38]. The area measures 50 mm (forward and backward) by 23 mm (upward and downward). In addition, to ensure user convenience when sitting, the exoskeleton on one side of the wheelchair board is removable, as shown in Figure 2b, red arrow. After sitting on the wheelchair, the exoskeleton is adjusted back to its original place. This design reduces crank blocks, which was a problem reported by users of the previous prototype. The incline of the chair back is adjustable per user preference. In summary, the wheelchair robot mechanism is adaptive, user-friendly and is operationally intuitive.

Two load cell force sensors are affixed to the pedal to obtain more precise force information and control data. The pedal force sensors are custom-made by Bengbu Sensor System Engineering Co. Ltd. (Anhui, China) [39]. The foot-to-surface contact area is spherical such that it enables effective contact between the foot and force sensor across a broad range of directions and angles. The measuring range of the force sensor is 0-300 N with a sensitivity of 1.5±0.1 mv/V and accuracy of 0.1 % FS. Figure 2 shows the sensor pedal design with a zoomed view, where the installation area is specifically shaped to host the sensor.

Figure 3 shows the wheelchair robot control design. A sensor fusion method has been proposed for the motion control. The wheelchair locomotion and leg cycling motions are coordinated by force values provided by force sensors, as well as by the joystick on the operation panel. In addition, the information of the position sensors of the motors are also considered in the proposed PD control. Force sensors affixed to the pedal detect the pedal force, which is then used to acquire data for user leg cycling motion intention. In this paper, the intended motion of the user is treated as a defined leg cycling exercise motion intention, as shown in Figure 3. The wheelchair locomotion and leg cycling motions are coordinated by force values provided by force sensors as well as motion choices via the operation panel. When the user motion intention is “no”, then the wheelchair uses motion commands and is propelled forward by the motors only. Here, the user may feel that the wheelchair motion is self-actuated, which can create a greater sense of safety. Under the present system, the detectable user cycling motion intention includes the presence or absence of leg cycling and wheelchair motions. There are three system modes to assist user leg movement: passive, active and active-passive. In the operation panel, the rotary knob can be adjusted to wheelchair motion, leg motion or both motions (i.e., wheelchair and leg). Once leg motion commences, as shown in Figure 3, “yes” option, then the assistance mode is determined by the detected pedal force, i.e., it is adjusted to match the ‘fitness’ level of the user. When a user sits on the wheelchair, he or she is asked to attempt one full cycle; the force sensor then records the force to determine the physical condition of the user. If the user does not have enough strength to independently cycle, then the passive leg motion mode is advised. Alternatively, the user can attempt independent cycling with the force sensor $F_{f_{0}}$ recording the pedal force. The reference force, $F_{f_{0}}$, is calculated by Equation (1). The motion mode is given by the values of Δ*F_m_*(*t*), as defined in Equation (1). Here, if Δ*F_m_*(*t*) is the largest value, then the active mode is engaged; if 0 < Δ*F_m_*(*t*) < $F_{f_{0}}$, then the active-passive mode is engaged. Thus, the leg-exoskeleton is propelled either by the legs of the user or by the motor.

Based on force values at different cycle positions between the left and right pedals, the trajectory control can be planned for well-regulated operation, which is required to apply force information to motion control. The control system records force values every 50 ms during the cycling motion via force sensors mounted on the pedals. Once 20 values are recorded (chronological order), then the system computes an average value to be used as the input force for the control; the force valuation is expressed by:(1)$$\Delta F_{m}\left(t\right)=\frac{\sum_{i=1}^{i=20}F_{i}}{20},$$ where the force variation, Δ*F_m_*(*t*), is the force value computed by the 20 values of *Fi_i_* obtained from the force sensors with acquisition intervals of 50 ms. This calculation is repeated during the cycling operation. Once the average force values of the left and right pedals are obtained from Equation (1), then the higher value is chosen for the motion control.

The control design is presented in terms of a user-oriented solution. Figure 4 shows the control design scheme. The control system has two inputs: *F_m_*(*t*) computed from the force sensors and *F_j_*(*α*) computed from the joystick signals. The STM32F4M4 is the microprocessor used. The voltage signal, $u_{f_{\left(t\right)}}$, obtained from the force sensor, is computed as a function of *F_m_*(*t*); $u_{f_{\left(t\right)}}$, obtained from the joystick, is computed as a function of *F_j_*(*α*). The values are used as input data for the control algorithm, as shown in Figure 4. *M_l_* represents the left motor; *M_r_* represents the right motor. *P_j_* is the proportional control from joystick commands to operate the motors *M_l_* and *M_r_* of the wheelchair. *G*_(*s*)_ is a model for the operation of the leg-exoskeleton motor. A PD feedback controller controls the velocities of each motor by generating PWM signals. The D control feature is computed from the motor position acquired from an encoder with *K_j_* = 0.2. The control P feature is identified by calibrating with *K_p_* = 6.9. The feedback from the leg-exoskeleton serve-motor, is combined with $u_{f_{\left(t\right)}}$ to give the control signal, *u_e_*_(*t*)_. Gains *K_j_* and *K_p_* along with encoded response *d_θ_*_(*t*)_/*d_t_* generate PWM commands for the motors. The system output is wheelchair motion velocities, ν*_w_*_1(*t*)_, ν*_w_*_2(*t*)_, and the cycling motion speed, ν*_b_*
_(*t*)_. Furthermore, the wheelchair velocity can be adjusted per user preference. The PD control is intentionally fairly simple, in order to satisfy market needs (i.e., a ‘value-add’ against competing products) and to ensure user-oriented operation.

To analyze the PD control system performance, a simulation is performed in Simulink with a control design scheme. In the simulation, per the servo-motor parameters for the leg exoskeleton, the Laplace transform function can be expressed as: G(s)=Kas+b, with *K* = 65, a = 6.9, b = 1.2, *K_d_* = 0.2. The simulation results show that the PD control system runs well with a response time of 0.17 s. The processing-speed of the data analysis is 1 KHz due to the processor chips.

Per the control design scheme, as shown in Figure 4, and block algebra rules, the output and input speeds can be expressed as:(2)$$\frac{v_{b}\left(t\right)}{u_{e}\left(t\right)}=\frac{\left(K_{p}+\frac{d_{\theta (t)}}{d_{t}}\right)G\left(t\right)}{1+\left(K_{p}+\frac{d_{\theta (t)}}{d_{t}}\right)G\left(t\right)},$$
(3)$$\frac{v_{w}\left(t\right)}{u_{j}\left(t\right)}=K_{j},$$
where $u_{e(t)}=u_{f(t)}-u_{b(t)}$. The output velocities (from Equations (2) and (3)) are regulated and coordinated with the force sensor feedback and input signals from the control panel. 

## 4. Prototype and Experimental Testing

Figure 5 shows another prototype built by the Intelligent Robot Institute (IRI) at the Beijing Institute of Technology (BIT). The prototype dimensions are 910 mm × 590 mm × 1037 mm. While the width is similar to that of a standard wheelchair, the length is longer due to the pedaling system. The turning circle radius is approximately 550 mm, which allows the cycling motion to be engaged in areas exceeding 2 m^2^. If the space inside the home is insufficient, then the user can exercise outdoors or can pedal with the wheelchair in the stationary position, similar to using a treadmill in a home environment. The operation panel has function buttons for motion modes, speed adjustments and emergency stoppage. The different functions allow the wheelchair motion to be adjusted to the specific physical needs of each user.

The prototype has leg exoskeletons and a pedal system attached to a wheelchair. The leg exoskeleton structure is made from polytef with adjustable links for customization based on user size. The motor was manufactured by the Huisitong Company (Beijing, China). The motor torque of the exoskeleton is 0.27 Nm. To generate the power needed for a normal cycling motion, a planetary reducer is selected with a reduction ratio of 47:1; the motor system output is approximately 13 Nm. The pedal unit is driven by a gear-gear stripe transmission system. The overall weight of the prototype is approximately 25 kg, including the wheelchair structure. The weight of the leg exoskeleton structure is approximately 2 kg.

Experiments were conducted with fully-abled volunteers and aimed to: determine the feasibility of the mechanical and control designs; characterize operations of the prototype; and evaluate the effectiveness of the exercise per data obtained from the EMG sensors. The experiments were approved by the ethical committee of Peking University with consent obtained from the College of Engineering, Peking University, under approval number IRB2014-02-02.

### 4.1. Feasibility Experiment

The experiment only recorded information from the force sensors when the crank was engaged. Figure 6 shows different stages of one complete pedaling cycle using the crank. The pedaling experiment only examined the active motion mode. The experiment used a wheelchair with an affixed leg-exoskeleton that was manually propelled by the volunteer. After a physical warm-up, the volunteer was asked to cycle at a fixed speed for a duration of two minutes. The force values were recorded for the duration of this time.

Figure 7 shows the acquired force values, with plot-points indicating a typical cycling motion. In one leg, there are different force values at different pedal positions of the cycling motion [40]. The force values periodically change (approximately every 2 s) throughout the pedaling cycle. The mean force value is approximately 70 N; this value is the leg weight of the user on the pedals. In the experiment, the force values of the left leg ranges from 38 to 95 N and the right leg ranges from 38 to 130 N, which indicates that the right leg of the user is stronger than the left leg. The force values are used as the initial input control signals for *F_m_*(*t*).

### 4.2. Muscle Exercise Experiment

The experiment evaluated the effectiveness of exoskeleton-based exercise per the recorded EMG signals at the left and right gluteus medius muscles. The gluteus medius muscle was tested because of its important role in body balance. If the proposed wheelchair can provide effective exercise for this muscle, then it is a viable health improvement technology. To demonstrate the effectiveness of exercise under robot mechatronics, the experiment examined both walking and sitting pedaling modes for three fully-abled persons. Walking and cycling are two exercise modes for the lower limb muscles. These exercises prevent muscular atrophy and recover walking ability. This paper compares cycling and walking exercises using the same EMG activation reference. Even though the exercises are different, a comparison of the results still shows the effects of the exercises, thereby validating the comparison. Here, walking was done in the absence of the wheelchair. Figure 8 shows the EMG sensor positions at the left and right gluteus medius muscles.

The EMG dataset included data collected from users. A custom-built, 2-channel EMG acquisition system was used to acquire the data at a sampling rate of 2000 Hz from the left and right gluteus medius muscles. MATLAB® 2016b software (Mathworks, Natick, MA, USA) performed the analyses. A reference electrode was placed dorsal to the vertebration position, where the muscles are weaker. Raw signals from the reference electrode and intramuscular electrodes were received.

Three persons with different statures (strong, middle and thin) are executed to the experiments, whose basic parameters of the statures are shown in Table 1. Subject selection is determined by the diffidence of the gluteus medius muscles. Even though the body sizes of the three test subjects were similar, their gluteus medius muscles had different characteristics. These characteristics were divided into "strong", "middle" and "thin". Each subject was trained in robot operation and experimental procedures. 

The experimental protocol included three steps. For each volunteer, the first and second steps consisted of a series of six walking and active pedaling tests at a comfortable self-selected speeds for approximately 60 seconds. The experiments were repeated six times and the optimal results for each subject was selected. Tests were repeated if they exceeded ±5 % of the average walking or pedaling speed as established during the warm-up phase. The third step consisted of a series of passive cycling at different speeds when exercising with the wheelchair. The actuated exoskeletons supplied the driving force to ensure pedal speeds of 16, 32, 48, 64 and 80 r/min for approximately 60 seconds during each test. The EMG voltage values were recorded during the process for subsequent analysis of muscle reactions. To analyze the reaction of the gluteus medius muscle during testing, the frequencies of the EMG voltage values for the activated muscles in one rotation are calculated based on the average value, given by:(4)φMa=∑i=1i=nuinu0×100%,
where *n* is the times of muscle activation, and *u_i_* is the value of the EMG voltage and *u*_0_ is the reference maximum voltage value of activated muscle [41]. Thus, Equation (4) can evaluate the percentage of activated muscles during a test. Statistic results for this evaluation are shown in Figure 9. 

From Figure 9, it can be observed that the gluteus medius muscles are activated both in walking and cycling in different speeds. During the walking, the computed percentages are close to 90%, indicating that the muscles are almost fully activated for all volunteers. During cycling motions, the percentages are measured near to 75% when cycling at 16 r/min, while the percentages are approximately 85% when cycling at 32 r/min. Thus the frequency of muscle activations during walking is higher than during cycling. In addition, percentages from three volunteers when cycling at 16 r/min show that different body sizes present different muscle activations. Volunteers A gets highest percentage indicating that strong persons can get better exercise than thin persons in cycling motion. The other volunteers present similar results and the results from volunteer B are selected as a typical example to analyze the characteristics of the exoskeleton-assisted wheelchair.

Raw EMG signals passed through a differential amplifier at a gain of 1000. Here, an analog filter with a band-pass width of 20–2000 Hz was selected. The present design sampled the relative narrow center frequencies (i.e., 180–220 Hz). The sampling frequency was 2000 Hz. To remove the low frequency movement effect and to confirm minimal interruption to the raw EMG signal, a high-pass 4th order Butterworth filter with phase lag was applied (with a cut-off frequency of 40 Hz). EMG data can evaluate successive events independently or treat them as single event; here, the ‘event’ represents the presence or absence of muscle stimulation. For the present experiment, events that occurred less than 300 ms apart were treated as a single event. Per [42], the envelope method is commonly used to process EMG data. However, the present experiment aims to detect and characterize numerous EMG events over a longer recording period such that the filtered and rectified signals are integrated into an output; as such, the phasic level events were utilized. The phasic activity in an EMG signal was detected and parameterized. The minimum amplitude of the phasic electro-muscular activity was set and the signal was analyzed. Each change that exceeded the given limit was recorded as an active event. For each muscle (left and right) of the user, an ensemble average was generated with a threshold of 100 mV for each active event. All user ensemble averages were summed and re-averaged to produce a total ensemble for each event, which was then used to establish an EMG profile for each movement across the gait or cycling cycle. In addition, the maximum amplitude of each event was also recorded. The accuracy and repeatability of signals from the EMG sensors were checked at the calibration phases. The characteristic accuracy error was 3 % (approximately 10 mV) and the repeatability error was 0.5 % (approximately 1.6 mV).

Data was recorded for activity on the posterior segments of both left and right sides of the gluteus medius muscles while the users were walking and pedaling in the active and passive modes. After being filtered and rectified, muscle activities showed similar patterns and symmetric waves for both left and right sides when walking and pedaling, as shown in Figure 10, Figure 11 and Figure 12. The EMG signals for minimum activity, mean integral and peak amplitude were used as references for the effectiveness of the exercise, the peak amplitude of the EMG signals for each of the three movement conditions ranged from 1000 to 2000 mV. The amplitude verifies the effectiveness of the muscle exercise, while muscle activity during walking was more pronounced and fluctuating, hence, muscle activity during pedaling is able to achieve effective muscle exercise through EMG signal based stimulation. The fluctuating muscle activity that occurs during walking means that the cycling motion requires more time than the walking motion to achieve the same degree of exercise effectives. Also, the right gluteus medius muscles are much stronger on the right side than the left side; here, the amplitude of the left muscles are approximately 1000 mV, while the right is approximately 1600 mV. Figure 10, Figure 11 and Figure 12 show that the motion has symmetry characteristics for the left and right sides.

To further clarify the effectiveness of the wheelchair robot exercise, the amplitude characteristics of the EMG signals are analyzed, as shown in Figure 13. A comparison of mean and maximum values for each active event during walking and pedaling in the passive mode (at 16 r/min) shows that the degree of muscle stimulation is equal to or lower than that of the walking process for pedaling with exoskeleton support; in addition, the movements are still less active than that of standard walking. Specifically, the mean values in passive cycling mode at left side are around 240 mV, while the amplitudes are around 300 mV during walking. In addition, the cycling mode shows bigger fluctuation than walking, which indicates gluteus medius muscles can get more activated during cycling. For the maximum values, the variation range of cycling is from about 800 to 1600 mV, while the walking motion is from about 1000 to 1800 mV.

To determine the exercise effectiveness at different cycling speeds, the mean and maximum EMG signal values at the left gluteus medius muscle locations for passive pedaling at different speeds were recorded, as shown in Figure 14. Here, some of the muscle activity mean and maximum values increase as the pedaling speed increases, although the rise is not significant. For the mean value, the maximum amplitude appears at the speed of 80 r/min. At a speed of 48 r/min, the muscles can get the minimum activation. For the mean value, the muscles activations are almost the same at the speeds of 16 r/min and 64 r/min, while the maximum value has a big difference. Consequently, choose the mean or maximum value to evaluate the muscle exercise is important in different cycling motions.

Table 2 lists EMG signals recorded at 9 s across the different motion modes and speeds. For greater clarity, the standard deviation values are included with the results. The quantitative results are discussed and analyzed based on these values, a comparison of the mean and maximum values of the EMG signals can validate the exercise effectiveness of the present design, as shown in Table 2. In the walking mode, the mean value is 280 mV, while the minimum value is 240 mV for different speeds in the passive cycling mode. Consequently, the results show that for abductor muscles, the cycling efficiency at different speeds (from 16 to 80 r/min) in the present wheelchair is marginally lower than that of walking at 3 m/s. This shows that passive cycling in the present wheelchair robot can effectively exercises user legs to the same extent as slow walking. From the maximum values, the same result can be deduced. In addition, a comparison of the different passive cycling modes shows that faster speeds do not guarantee more effective muscle exercise. This phenomenon can be explained by cycling motion characteristics [40]. At 16 r/min, the mean value is 220 mV, while at 32 and 80 r/min, the mean values are 240 mV. For 64 r/min, the highest mean and maximum values are obtained for the current design, i.e., the optimal speed for maximum cycling exercise effectiveness. 

Scientifically, a proper statistical significance of testing a set of subjects and repetitions is needed, but the abovementioned aim of the paper is satisfied by reporting the results with one subject after several experiments. Testing with patients will require medical protocols to be developed for future work on improvement of the proposed design, also with clinical considerations. For elderly and stroke experiment participants, the muscle activation characteristics would likely approximate that of the healthy volunteers, especially for weakened or injured muscles. The immediate benefit of the design is that a user is able to move around while simultaneously exercising leg muscles during the cycling motion, which is good for activating leg muscles and restoring independent walking.

## 5. Conclusions and Future Work

This paper presents a mechanical design and control method for an assistive wheelchair robot that promotes locomotion and exercises the gluteus medius muscles using a wheelchair with attached exoskeletons. The mechanical design and control system characterize the operation of the wheelchair robot under several ‘daily-routine’ functioning conditions per the leg action of the user. Experimentation with a prototype shows the feasibility of the present design, which combines leg exoskeletons with a wheelchair to exercise the gluteus medius muscles of the user. In addition, EMG signal testing during walking, passive pedaling and active pedaling confirms that the wheelchair robot effectively exercises the gluteus medius muscles.

Future work will undertake further experimentation to confirm the feasibility of the mechanical design and master-slave control. The current work only engaged fully-abled users for the experiment; future work should also engage users with different levels of lower limb fitness. Furthermore, the present work only used the EMG signals for the gluteus medius muscle to determine the effectiveness of the exercise; to further evaluate the integrated effectiveness of the muscle exercise, future work should also incorporate leg muscles such as quadriceps and the hamstrings.

## 6. Patents

[1] Ceccarelli M., Huang Q., Huang G., Wheelchair with exoskeleton for assistance of leg motion, Italin Pat. No. 102015000032950, 12 November 2017.

[2] Huang Q., Liu J., Huang G., Zhang W., Yu Z., Chen X., Meng F., Liu H., An exoskeleton robot for rehabilitation and movement based on master-slave control, China Patent No.: CN201610948525.7, 26 October 2016 (in Chinese)

[3] Huang Q., Huang G., Ceccarelli M., Tian Y., Zhang W., Yu Z., Chen X., A pedal-actuated exoskeleton wheelchair for lower limb exercise, China Patent No.: ZL201510239279.3, 12 May 2015 (in Chinese)

## Figures and Tables

**Figure 1 sensors-19-00548-f001:**
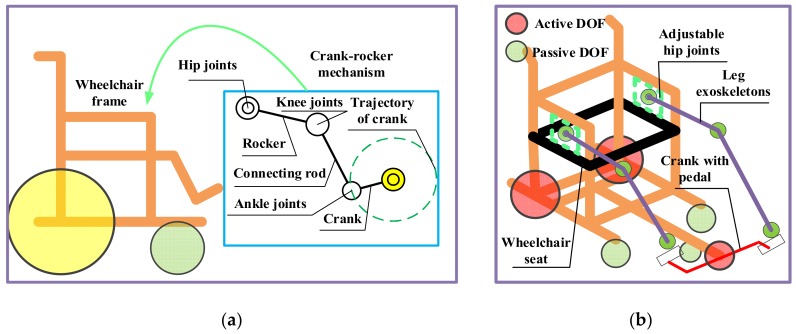
Conceptual design of wheelchair robot with leg exoskeletons: (**a**) Kinematic design; (**b**) DOF configuration.

**Figure 2 sensors-19-00548-f002:**
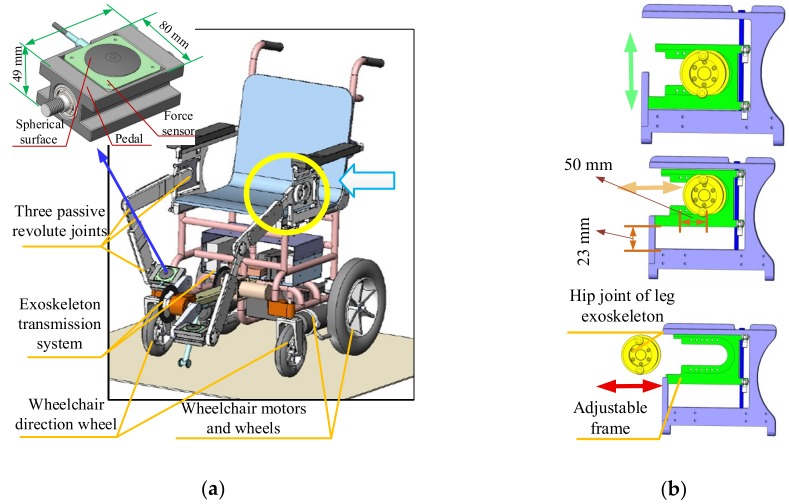
CAD design solution for novel leg-exoskeleton wheelchair robot: (**a**) Full mechanical design; (**b**) Hip joint adjusting mechanism.

**Figure 3 sensors-19-00548-f003:**
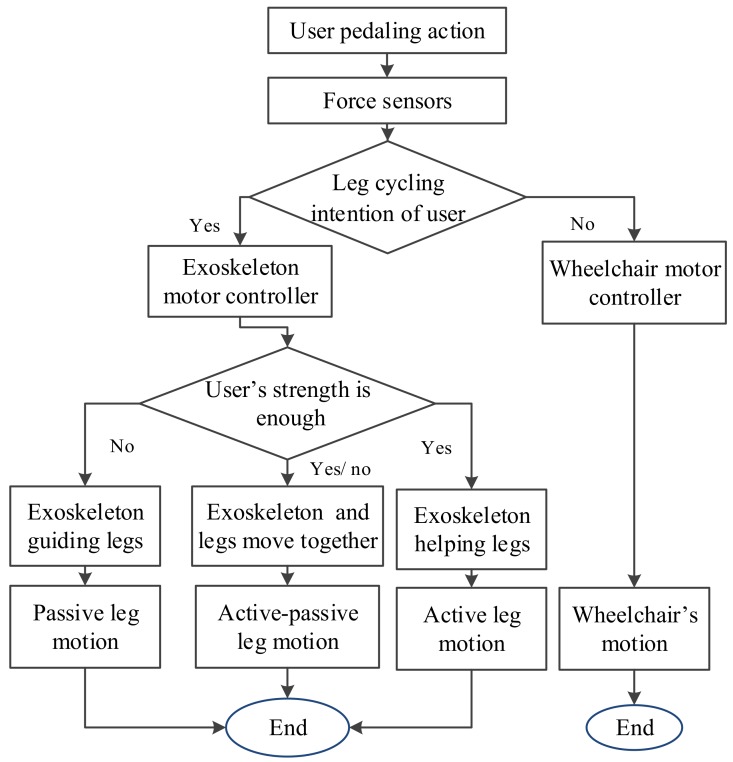
Control strategy flowchart for the wheelchair robot in Figure 1 and Figure 2.

**Figure 4 sensors-19-00548-f004:**
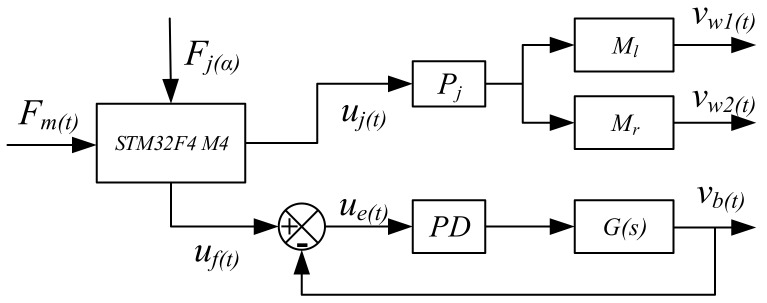
Control design scheme of leg exoskeleton assisted wheelchair.

**Figure 5 sensors-19-00548-f005:**
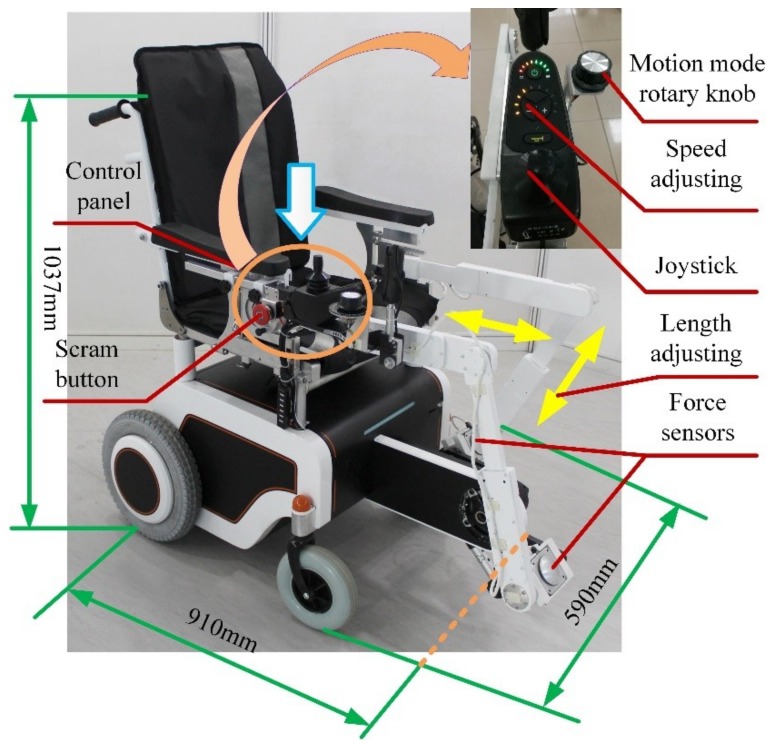
Newly developed wheelchair prototype with linked exoskeletons from the Intelligent Robot Institute at the Beijing Institute of Technology per designs in Figure 1 and Figure 2.

**Figure 6 sensors-19-00548-f006:**
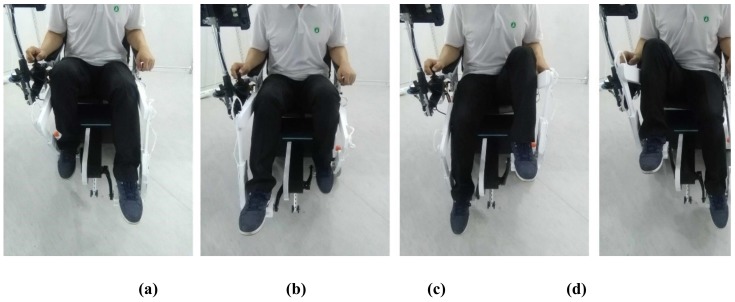
Still images from feasibility experiment during one complete cycle of the crank in different positions: (**a**) Top dead center; (**b**) Power phase; (**c**) Bottom dead center; (**d**) Recovery phase.

**Figure 7 sensors-19-00548-f007:**
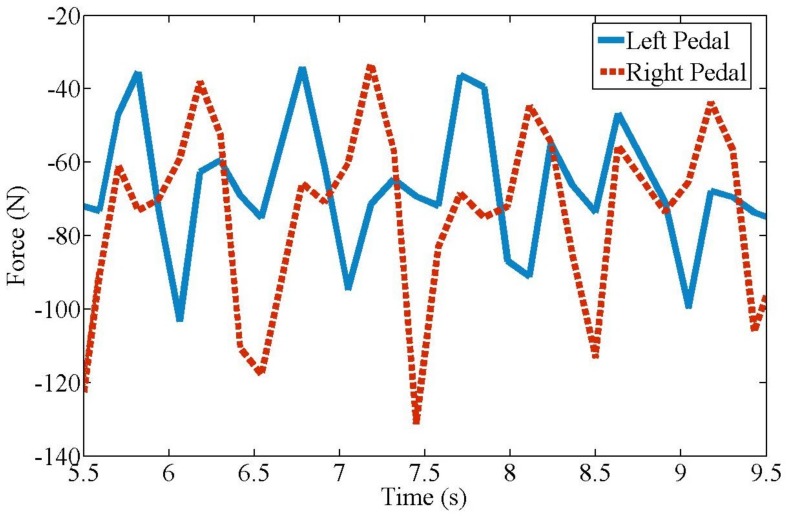
Forces on pedal during experimentation as shown in Figure 6.

**Figure 8 sensors-19-00548-f008:**
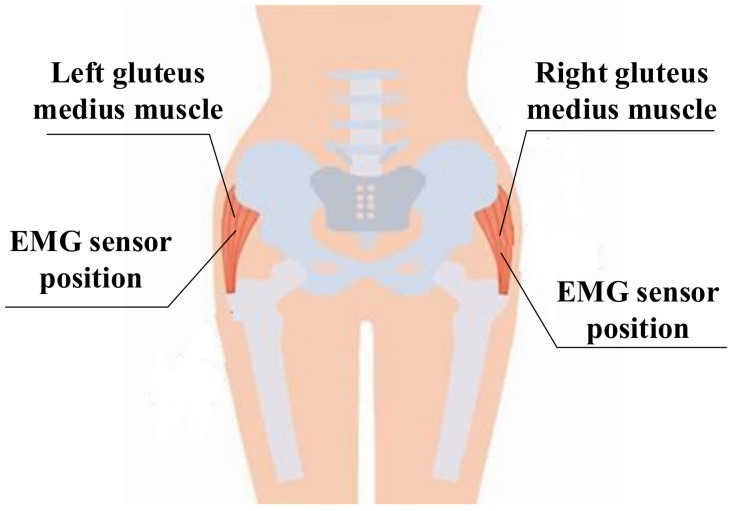
EMG sensor positions at left and right gluteus medius muscles locations.

**Figure 9 sensors-19-00548-f009:**
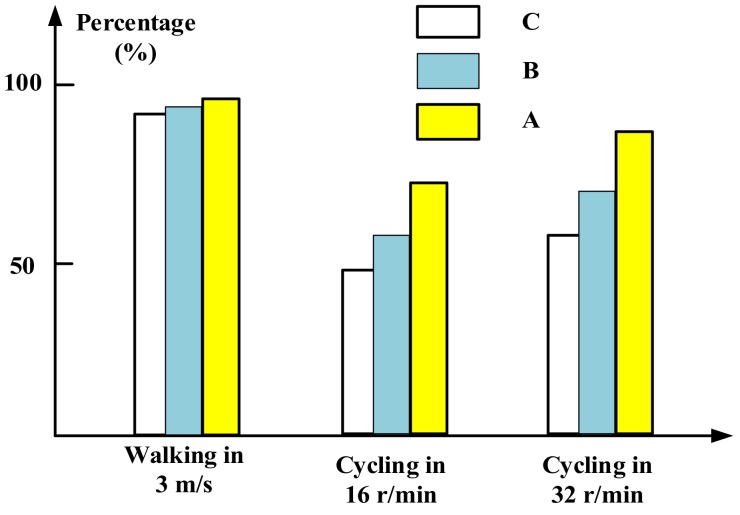
Frequency of activated EMG values during tests (Table 1).

**Figure 10 sensors-19-00548-f010:**
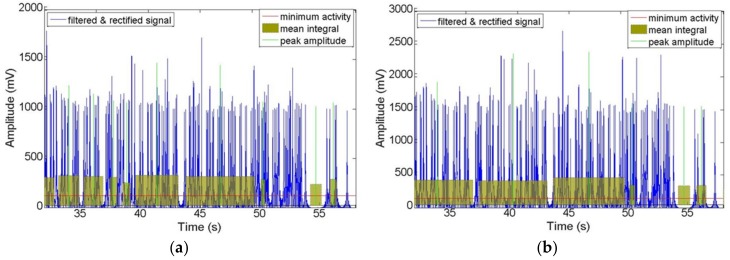
Recorded EMG signals at left and right sides of gluteus medius muscle locations during walking motion: (**a**) Left side; (**b**) Right side.

**Figure 11 sensors-19-00548-f011:**
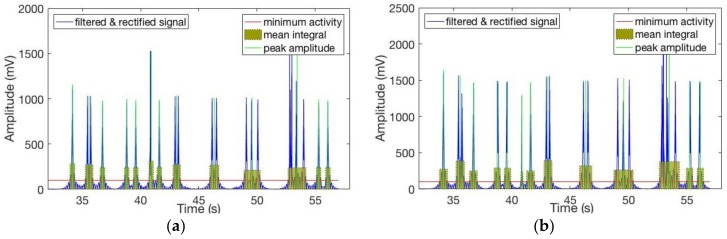
Recorded EMG signals at left and right sides of gluteus medius muscle locations during active cycling: (**a**) Left side; (**b**) Right side.

**Figure 12 sensors-19-00548-f012:**
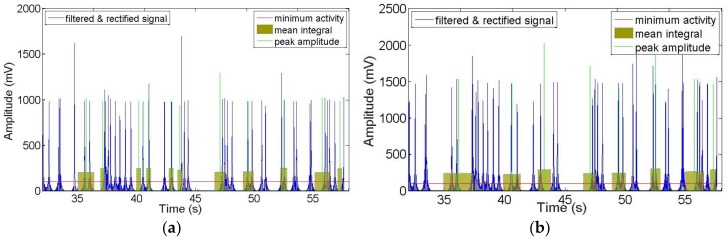
Recorded EMG signals at left and right sides of gluteus medius muscle locations during passive cycling: (**a**) Left side; (**b**) Right side.

**Figure 13 sensors-19-00548-f013:**
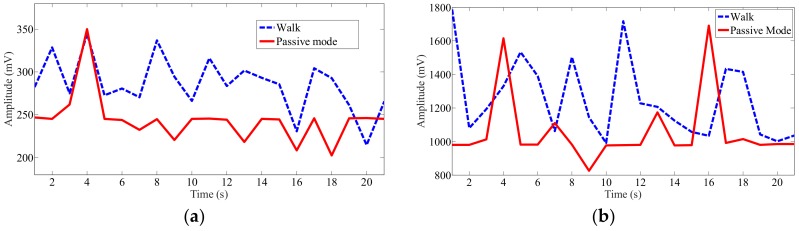
Mean and maximum recorded EMG signal values at left gluteus medius muscle location for each active event during walking and pedaling in passive mode at 16 r/min: (**a**) Mean value at left side; (**b**) Maximum value at left side.

**Figure 14 sensors-19-00548-f014:**
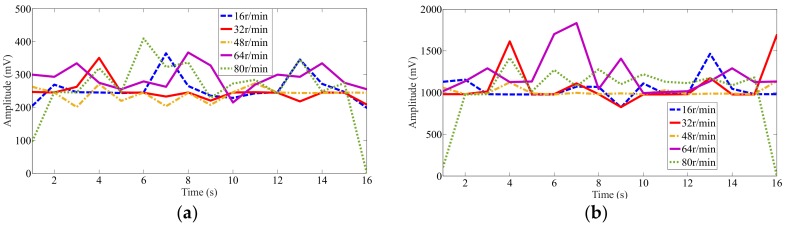
Mean and maximum recorded EMG signal values at left gluteus medius muscle locations for passive pedaling at different speeds: (**a**) Mean value at left side; (**b**) Maximum value at left side.

**Table 1 sensors-19-00548-t001:** The main parameters of the volunteers for the experiments.

Volunteers	Height (cm)	Weight (Kg)	Age
**A (Strong)**	165	56	30
**B (Middle)**	168	50	29
**C (Thin)**	171	52	28

**Table 2 sensors-19-00548-t002:** Recorded EMG signals at 9 s for different motion modes and speeds.

Modes	Speeds	Mean Value (mV)		Max Value (mV)	
		Value	Standard Deviation	Value	Standard Deviation
**Walking mode**	3 m/s	280	27.41	1150	480.81
**Passive cycling mode**	16 r/min	220	40.66	900	176.99
	32 r/min	240	37.41	900	184.90
	48 r/min	230	40.09	1000	89.90
	64 r/min	310	41.05	1200	281.75
	80 r/min	240	50.58	1100	294.49

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
