# Peer review of "Design and Feasibility Study of a Leg-exoskeleton Assistive Wheelchair Robot with Tests on Gluteus Medius Muscles"

_sensors, 2019, doi:10.3390/s19030548_

Reviewer 1 Report

Ethical concern :

There seems to be no informed consent or ethical approval obtained for the human subject data collection. In one place( Line 294) authors state data from one able-bodied participant was collected. However when they present the results there are three subjects data.

The authors have to clearly state if IRB approval was obtained, if so the approval number, IRB institution name , if informed consent was obtained. IF the institutional IRB waived the consent process for this study, the authors have to state that clearly declare that decision and provide the IRB reference number.

If IRB approval was not obtained, the authors cannot use human subject data collected for the manuscript.

Other major concerns:

Title : The authors test only gluteus medius however the title seems to generalize 'Lower limb muscle exercises' ( which seems to suggest they in\licude more muscle groups from lower limb and a series of exercise). This is an overstretch. Please rephrase title to state gluteus medius.

This is more of a design-feasibility study. this should be included in the title.

It is not clear what the authors mean by "integration to locomotion " in the title? How do they integrate it?

Study design major concerns:

The authors seem to compare walking Vs seated cycling for muscle activation. These are two very different tasks and like an apple-to-orange comparison. Further the authors claim that there is 9% more EMG activation while seated cycling than walking ( in abstract and main text) seems misleading as based on Figure (9) , this does not seem true ( Lines 327 -328 - the muscle activation in walking > cycling unlike the authors description). More detailed clarity in writing and results needed as to what the author wants to convey to the reader.

Individuals with stroke/disability may not be able to apply symmetric forces to pedal. Since the control system takes only the maximum of right/left forces,it is not clear if the individuals with disability will have to apply this higher force on their disabled side to pedal as well. This has not been addressed clearly even for healthy controls.

The device seems to add an extra 25 Kg weight to the wheelchair. So in the active mode, the participant have to peadal 25 Kg more weight in addition to their body weight. This may not be comfortable or safe ( may lead to fatigue or injuries later).

The mechanism is pitched to be working in-home environment. However, tere is a usability issue here, usually home space has lot of furniture and there is not enough space to pedal around. Also, no information on braking response is given. These aspects have to be discussed. The feasibility study have to through some light on implementation for in-home environment. From Lines 202 and 203, seems like the user will need a long open space in the house to use this. Often people live in studio apartment or care-homes, these usability issues have to be discussed

Also, such technology for self training have safety concerns. For individuals with disability, the physiologic condition ( heart rate and BP) before and after training have to be carefully monitored to find training dosage. Self administered training based on force at pedal may nlead to unsafe conditions. These aspects should be discussed.

The muscle activation were not normalized to MVC. The results trends may vary for MVC normalized muscle activation.

Line 378-379 : The authors have to show using statistics that indeed fluctuation over mean is higher for cycling mode. There is not data proof provided to support this statement.

Conclusion : Line 417 -418 : There is no evidence in ths work to support that this device (i) promotes locomotion or (ii) exercises all lower limb muscles. such claims are over stretch. This is a feasibility study and there is no evidence to support clinical efficacy yet.

Other concerns:

Were the force sensors FSR sensors or load cells?. Please provide more details. If FSR sensor, Line 192 should read a circular instead of "are is spherical".

It is not clear if the control system would choose the speed based on pedal force data . How does it modulate/change speed/rpm?Please explain these in detail

Reviewer 2 Report

The manuscript presents a very interesting and innovative design of a wheelchair with exoskeleton aimed to stimulate lower limb movements during wheelchair mobility.  The paper describes the conceptual design and ptototype features (mechanical and controlling system) with figures, which is really helpful for the readers to understand the principles and mechanisms proposed. Secondly, the authors present an experiment with subjects without disability, with the evaluation of EMG of the gluteus medius. Although the first part (conceptual design and prototype description) seems to be the most important part of the paper, the experiment with preliminary tests of the prototype contributes to improve the paper quality. However, in the experiment description, the authors should provide more information on the muscle investigated (gluteus medius). Although this is an important muscle in the control of hip and pelvis, there are other muscles that are more important to walking ability, such as gluteus maximus (that extend the hip and is very important in the stance phase of the gait), quadriceps femoral and the hamstrings. These three groups of muscles are very important in the gait cycle and, when they are affected, the effect on walking ability (and therefore the possibility of end up needing a wheelchair) is greater than the gluteus medius (although guteus medius plays an important role on the lateral stability of the pelvis). Therefore, it would be positive if the authors provided an explanation on why only gluteus medius muscle was assessed. Also, in the discussion of the results, I suggest the authors to make some statements on how they would expect the results (in terms of muscle activation - EMG) if the sample of subjects comprised elderly people and stroke patients, for example. What might be the immediate benefits and to what extent the use of this system would help the subjects to recover walking independence?

FInally, it would be positive if the authors shared their thoughts about how this system could be implemented in rehabilitation practice, that is, how it would compliment traditional practices in rehabilitation routine.

Reviewer 3 Report

This paper introduced unique rehabilitation method using  electric wheelchair.But this paper has some mistake, I do not understand this paper`s conclusion.

[1] Table 2, is a mistake? There is no evidence of 9% higher than walking.

[2] I do not understand the relationship between these results of Fig 11,12,13 and the results of Fig 13. It seems like the opposite result. Please add the explanation in more detail.

Author Response

Round  2

Reviewer 3 Report

This paper introduced unique rehabilitation method using  electric wheelchair.Please confirm the following comment
1)From Fig10(a) and Fig12(a), mean integral value of Fig 10 (a:walking) is larger than mean integral value of Fig 12 (a:cycling). However, mean value of walking is lower in Figure 13 (a).I think that this is related to the length of one task. If so, please show the time of one task average.Also please explain how to decide the range of integral value in the Fig 10,11,12.
